# Effects of *Quillaja* Saponin on Physicochemical Properties of Oil Bodies Recovered from Peony (*Paeonia ostii*) Seed Aqueous Extract at Different pH

**DOI:** 10.3390/foods12163017

**Published:** 2023-08-11

**Authors:** Pengkun Shen, Ruizhi Yang, Yingying Wu, Jiao Liu, Xiuzhen Ding, Wentao Wang, Luping Zhao

**Affiliations:** College of Food Science and Engineering, Shandong Agricultural University, Tai’an 271018, China; 17852262997@163.com (P.S.); ruizhiyang2001@163.com (R.Y.); yying1220@126.com (Y.W.); liujiao202202@163.com (J.L.); xzd@sdau.edu.cn (X.D.); wangwt@sdau.edu.cn (W.W.)

**Keywords:** peony oil body, extraction pH, *Quillaja* saponin, particle size, zeta potential, oxidative stability

## Abstract

Peony seeds, an important oil resource, have been attracting much attention because of α-linolenic acid. Oil bodies (OBs), naturally pre-emulsified oils, have great potential applications in the food industry. This study investigated the effects of extraction pH and *Quillaja* saponin (QS) on the physicochemical properties of peony oil body (POB) emulsions. POBs were extracted from raw peony milk at pH 4.0, 5.0, 6.0, and 7.0 (named pH 4.0-, 5.0-, 6.0-, and 7.0-POBs). All POBs contained extrinsic proteins and oleosins. The extrinsic proteins of pH 4.0- and pH 5.0-POB were 23 kDa and 38 kDa glycoproteins, the unknown proteins were 48 kDa and 60 kDa, while the 48 kDa and 38 kDa proteins were completely removed under the extraction condition of pH 6.0 and 7.0. The percentage of extrinsic proteins gradually decreased from 78.4% at pH 4.0-POB to 33.88% at pH 7.0-POB, while oleosin contents increased. The particle size and zeta potential of the POB emulsions decreased, whereas the oxidative stability, storage stability, and pI increased with the increasing extraction pH. QS (0.05~0.3%) increased the negative charges of all the POB emulsions, and 0.1% QS significantly improved the dispersion, storage, and the oxidative stability of the POB emulsions. This study provides guidance for selecting the proper conditions for the aqueous extraction of POBs and improving the stability of OB emulsions.

## 1. Introduction

Oil tree peony, an important woody oil crop in China, has a long history of cultivation of around a thousand years. The oil content of peony seeds is 18.30~24.04%, and they could produce healthy and edible vegetable oil [1]. The peony seed oil contains saturated fatty acids (6.98~8.56%) and unsaturated fatty acids (91.44~93.02%), among which α-linolenic acid accounts for more than 40% [2]. It was found that vegetable oil existed in the form of natural lipid-storing organelles called oil bodies in rapeseed, beans, and nuts [3]. Oil bodies (OBs) are mainly composed of triglycerides (TAGs, 94.2~98.2%), oleosins (0.3~3.0%), and phospholipids (PL, 0.6~2.0%) [4]. OBs have a TAG matrix core coated by one monolayer of phospholipids embedded with structural proteins. Oleosins play crucial roles in keeping the structural integrity of OBs and resisting some external environmental stresses [5]. Based on the above structural characteristics of OBs and peony seed oil, peony oil bodies (POBs) have a potential broad application in both the food and cosmetics industries.

OBs with different components (such as proteins, neutral lipids, and phospholipids) were extracted by varying the extraction pH. Sun et al. (2023) [6] extracted peanut OBs at pH 4.0, 7.5, and 11.0, with the moisture and fat contents gradually increasing with the increasing pH, while the protein content decreased from 1.47% to 1.01%. Moreover, different extraction pH also affected the properties of the OB emulsions. Chen et al. (2014) [7] found that the neutral lipid percentages of soybean OBs increased from 85.88% to 91.89% when increasing the extraction pH from 6.8 to 11.0, while the percentage of proteins and polar lipids decreased. It was found that peanut OB emulsions extracted at pH 7.5 and pH 4.0 had smaller particle sizes, higher interfacial viscoelasticity, and a better emulsification performance than peanut OB emulsions extracted at pH 11.0 [6]. At present, Zhao et al. (2020) [8] extracted POBs at pH 7.0, with the stability of the POB decreasing by adding 50~100 mM NaCl, but the POBs were still good at 30~90 °C. However, the influence of the different extraction pH on the composition and properties of POB emulsions has not been systematically studied. In addition, it is important that OB emulsions have a good physicochemical stability for their applications in food. Polysaccharides and polyphenols were usually used to improve the physicochemical properties of OB emulsions in recent years. Interestingly, saponin, a natural small molecule surfactant, has excellent properties in reducing the interfacial tension of oil and water due to its amphiphilicity and improved emulsion stability. It was found that the particle size of the emulsion prepared with soy lecithin and medium chain triglycerides decreased from 0.36 μm to 0.15 μm by increasing the *Quillaja* saponin (QS) content from 0.5% to 2.5% [9]. Zhu et al. (2019) [10] prepared stable nano-emulsions with 1% tea saponins and found that the tea saponins could reduce the interfacial tension from 24.42 mN/m to 4.8 mN/m, improving storage stability. Therefore, saponins are effective in improving the common emulsion stability. However, only Liu et al. (2022) [11] found that 0.1% and 1.0% camellia saponins effectively delayed the lipid oxidation of the camellia OB emulsion extracted at pH 7.0. More research needs to be carried out in order to realize the effects of saponins on OB emulsions.

In this study, we extracted POBs from peony seeds at pH 4.0, 5.0, 6.0, and 7.0, and their compositions and isoelectric points were determined. The effects of QS (0.05~0.3%, *w*/*w*) on the particle size, zeta potential, microstructure, storage stability, and oxidative stability on the different POB emulsions were researched. This will provide theoretical guidance for improving OB emulsions and POB applications in food.

## 2. Materials and Methods

### 2.1. Materials

The peony seeds (*Paeonia ostia*, 10 kg) were obtained from Shandong Seed Industry Group CO., Ltd. (Jinan, China). *Quillaja* saponin was purchased from Shanxi Haosen Biological Technology Co., Ltd. (Xi’an, China). Protein Markers (14.4~97.4 kDa) were bought from Beijing Solarbio Science and Technology Co., Ltd. (Beijing, China) Mercaptoethanol, Bromophenol blue, Coomasil bright blue, glacial acetic acid, isoctane, isopropyl alcohol, methanol, n-butanol, thiobarbituric acid, sodium azide, trichloroacetic acid, and trichloromethane were purchased from Sinopharm Chemical Reagent Co., Ltd. (Shanghai, China). All reagents were of analytical grade.

### 2.2. POBs Extracted at Different pH

The extraction method of the POBs was performed with reference to the previous method by Nikiforidis et al. (2009) [12]. The peony seed kernels were cleaned with deionized water, soaked with deionized water in a beaker, which was then placed in a refrigerator at 4 °C for 16 h. The soaked kernels were mixed with deionized water (kernels/deionized water = 1/7, *w*/*w*), and ground for 2 min by a blender (18,000 r/min; Blst4090-073, Qster, Qingdao, China). The slurry was filtered using four layers of gauze, and the raw peony milk was collected, which was then adjusted to the required pH 4.0, 5.0, 6.0, 7.0, and 11.0. The mixtures were centrifuged at 13,433× *g* (4 °C, 30 min). The precipitate extracted at pH 4.0 was collected from the bottom of centrifugal tube; however, the floats extracted at pH 5.0, 6.0, 7.0, and 11.0 were collected from the upper layer. The collections were dispersed into DI water again, and the pH of the mixtures were adjusted to 4.0, 5.0, 6.0, 7.0, and 11.0. The mixtures were centrifuged in the same conditions as above. The precipitate extracted at pH 4.0 was collected and named pH 4.0-POB, the floats extracted at pH 5.0, 6.0, 7.0, and 11.0 were collected and named pH 5.0-, 6.0-, 7.0, 11.0-POB, respectively.

### 2.3. Measurement of POB Composition

The moisture content was measured using the oven-drying method, and the temperature was controlled at 105 °C. The measurement of the protein content was based on the method of Pan et al. (2022) [13]. Here, the coefficient of protein converted by nitrogen was 6.25. The lipid content was determined using Soxhlet extraction [14].

### 2.4. Tricine-SDS-PAGE

Samples were mixed with the sample loading buffer (containing 0.1% (*v*/*v*) bromophenol blue and 2% (*v*/*v*) mercaptoethanol). The protein concentrations of all samples were adjusted to 1.5 mg/mL via mixing with sample loading buffer. The mixed solutions were heated using a boiling water bath for 5 min, and they were centrifuged at 10,000× *g* for 5 min. Then, 10 μL of each sample was injected into the sample well. Electrophoresis was performed using a Mini-Protean instrument (DYY-8C, Beijing Liuyi Biotechnology Co., Ltd., Bejing, China) at a constant voltage of 30 mV until the mixed solution entered into the stacked gel; then, the voltage was changed to 100 mV until complete. The gel was fixed with 50% (*v*/*v*) methanol and 10% (*v*/*v*) acetic acid, and they were stained with 0.025% (*w*/*w*) Coomassie blue and de-colorized with 10% (*v*/*v*) acetic acid. The Image Lab 3.0 Software (Bio-Rad, Hercules, CA, USA) was used to take photos and analyze the intensities of the different protein bands.

### 2.5. Emulsion Preparation

The pH 4.0-, 5.0-, 6.0-, and 7.0-POBs were dispersed into deionized water to obtain the POB emulsions (10%, *w*/*w*), and their pH were adjusted to 7.0, respectively, using 0.1~2 M NaOH solutions. Each POB emulsion was, respectively, added QS (0.05%, 0.1%, 0.2%, 0.3%; *w*/*w*), and all samples were mixed well. To prevent microbial growth, 0.02% (*w*/*v*) sodium azide was added into all POB emulsions and the follow-up analyses were conducted immediately.

### 2.6. Zeta Potential 

The pHs of the different POB emulsions (10%, *w*/*w*) were, respectively, adjusted to 3.0, 4.0, 5.0, 6.0, 7.0 with 0.01 M NaOH and 0.01 M HCl solutions, and they were diluted 400 times with same pH buffers (phosphate buffer and citric acid buffer; 20 mM). Zeta potentials were determined using a Laser Nanometer Particle Size Analyzer (Malvern Instruments Ltd., London, UK) at 25 °C.

### 2.7. Particle Size

All POB emulsions were diluted 1000 times using a phosphate-buffered solution (20 mM, pH 7.0), respectively. The particle sizes were determined using a Laser Nanometer Particle Size Analyzer (Malvern Instruments Ltd., London, UK) at 25 °C.

### 2.8. Microstructures

Microstructures were observed and images were recorded at 25 °C using an optical microscope (MODEL BX-51, TKO Optical Instruments Co., Ltd., Tokyo, Japan). Samples (10 μL) were deposited on a microscope slide and covered with a cover slip. Samples were observed with a 10× eyepiece and a 100× objective lens.

### 2.9. Determination of Oxidative Stability 

Fresh 10% POB emulsions (pH 7.0), were stored at 37 °C with restricted lighting in closed glass vials for different time periods up to 14 days according to the method of Zhao et al. (2016) [15].

#### 2.9.1. Lipid Hydroperoxides

The peroxide value (POV) was determined using the modified methods described by Salminen et al. (2014) [16]. Each POB emulsion (0.3 mL) was mixed with 1.5 mL isoctane–isopropanol solution (*v*/*v* = 2/1). Each mixture was shaken twice for 20 s, and it was centrifuged at 900× *g* for 10 min. The supernatant (0.2 mL) was mixed with KSCN solution (20 μL, 3.94 M) and Fe2+ solution (20 μL, 0.072 M), and the volume was made to 10 mL with methanol–n-butanol solution (*v*/*v* = 2/1), which was kept in the dark for 20 min. The absorbance was measured at 510 nm using a UV-5100 spectrophotometer (Shanghai Yuan Analysis Instrument Co., Ltd., Shanghai, China), and the methanol–n-butanol solution was used as control. All peroxide values were calculated according to the standard curve using the following formula:$$\mathrm{POV}(\mathrm{mmol}/\mathrm{kg})=\frac{{M\times A}_{510}}{55.84\times 2\times m_{0}}$$

A_510_ represents the absorbance value of 510 nm; M represents the slope of the Fe^3+^ standard curve; m_0_ represents the colorimetric equivalent sample mass, g.

#### 2.9.2. Thiobarbituric Acid Reactive Substances (TBARS)

The secondary products of lipid oxidation were evaluated by TBARS according to the method by Jiang et al. (2022) [17], albeit with modifications. Thiobarbituric acid solution (TBA) was prepared by dissolving thiobarbituric acid (93.75 mg) and trichloroacetic acid (3.75 g) into 25 mL HCl (0.25 M). An amount of 0.5 mL of each emulsion was mixed with TBA reagent (1 mL) in a test tube, heated in a boiling water bath for 15 min, and cooled to room temperature. The mixture was added into 0.3 mL of trichloromethane, and it was centrifuged at 3000× *g* for 15 min. The supernatant was collected and the absorbance was measured at 532 nm. The TBARS content was calculated using a standard curve prepared from 1, 1, 3, 3-tetroxypropane (0~30 μM).

### 2.10. Calculation and Statistical Analysis

All POB emulsions were prepared three times on different days. Values of the different parameters were presented as the mean ± standard deviations. Mean and standard deviations were obtained by Origin Pro 2016b (OriginLab Co., Northampton, MA, USA). Statistical analysis was performed using SPSS statistics version with 25 ANOVA (IBM Inc., Armonk, NY, USA). The means were compared by a least significant difference (LSD) test at the 5% level (*p* = 0.05).

## 3. Results and Discussion

### 3.1. Component Analysis of POBs 

Table 1 showed that the contents of moisture, protein, and lipid of pH 4.0-, 5.0-, 6.0-, and 7.0-POBs. The pH 4.0-POB contained the highest moisture content (52.39%), followed by pH 5.0-POB (46.14%) and pH 6.0-POB (28.75%), and pH 7.0-POB had the lowest moisture content (20.98%). The protein contents of the different POBs decreased from 33.40% to 4.27%, while the lipid contents increased from 57.60% to 86.31% with increasing extraction pH from 4.0 to 7.0. This pattern was consistent with peanut OBs and soybean OBs extracted at pH 6.8~11.0 [15,18]. Considering the ability of proteins to bind water, it was believed that the water contents of the different OBs increased due to the increasing of protein contents [19].

Figure 1A showed the protein composition of POBs, and the protein types gradually decreased with increasing extraction pH. In order to determine the oleosins, we extracted the high purity pH 11.0-POB. Ishii et al. (2021) [20] reported that pH 11.0 was a simple method for obtaining high purity OBs. It was found that pH 11.0-POB contained three proteins (15 kDa, 17 kDa and 23 kDa). According to Zhao et al. (2020) [8] and Meng et al. (2020) [21], there were oleosin 15.06 kDa and oleosin 17.5 kDa in peony seeds. Therefore, 15 kDa and 17 kDa in electrophoretic gel were probable oleosins. In addition, Gao et al. (2018) [22] found that peony seed storage proteins contained 60 kDa, 48 kDa, 38 kDa and 23 kDa, respectively, and 38 kDa and 23 kDa were glycoproteins with isoelectric points (pIs) of 3.6 and 9.0. Zhao et al. (2016) [15] reported that storage proteins were released from protein storage vacuoles during grounding seeds, and some of them were adsorbed to the surface of soybean OBs, which were called extrinsic proteins of soybean OBs. Therefore, all POBs contained not only the oleosins (15 and 17 kDa) but also the extrinsic proteins. Both pH 4.0- and pH 5.0-POB contained four extrinsic proteins (23 kDa and 38 kDa glycoproteins, unknown proteins 48 kDa and 60 kDa), while 38 kDa glycoprotein and unknown 48 kDa were completely removed when the extraction pH was more than 6.0. This was probably because the extraction pH 6.0 and 7.0 were much higher than the pI 3.6 of 38 kDa glycoprotein, and the electrostatic repulsion between POBs and 38 kDa glycoprotein increased, which resulted in 38 kDa glycoprotein releasing from POBs. Interestingly, 5.60% 23 kDa glycoprotein was still bounded to pH 11.0-POB. It was attributed to the weaker electrostatic repulsion between POBs and glycoprotein 23 kDa (pI 9.0), and it is difficult to completely remove 23 kDa glycoprotein. Similarly, 11S globulin was removed from sesame OBs at extraction pH 8.0, whereas 2S albumin was completely removed at the extraction pH 11.0 [23]. It was likely that the interaction forces between the different extrinsic proteins and OBs were different. Zhao et al. (2013) [24] reported that the extrinsic Gly m Bd 30K and P34 probable thiol protease were strongly bound to soybean OBs through the disulfide bond with 24 kDa oleosin, and Gly m Bd 30K and P34 probable thiol protease were difficult to remove from soybean OBs. The analysis of protein band strength showed that the extrinsic proteins gradually decreased with increasing extraction pH from 4.0 to 7.0 (Figure 1B). The percentage of the extrinsic proteins gradually reduced from 78.4% of pH 4.0-POB to 33.88% of pH 7.0-POB, while the oleosin contents increased.

### 3.2. Zeta Potential 

The surface charges of the POBs were affected by environmental pH. As the environmental pH increased, the surface charges of the POBs decreased gradually from positive at pH 3.0~pH 4.0 to negative at pH 5.0~pH 7.0 (Figure 2A). The droplet charges were 0 at certain pH, which were the pI. The pIs of the pH 4.0-, 5.0-, 6.0-, and 7.0-POB were approximately 4.2, 4.3, 4.5, and 4.6, respectively. The pIs of POBs showed an increasing tendency with increasing extraction pH, which is consistent with the results reported by Zhao et al. (2016) [23]. In addition, pIs of pH 4.0- and pH 5.0-POB were close to their extraction pH, which resulted in more extrinsic proteins bounding to pH 4.0- and pH 5.0-POBs with the weaker electrical repulsions. This explained why the extrinsic protein of POBs gradually decreased with increasing extraction pH from 4.0 to 7.0 (Figure 1).

Figure 2B showed the effect of QS on the zeta potential of the different POB emulsions (pH 7.0). The zeta potential of the pH 4.0, 5.0-, 6.0, and 7.0-POB emulsions without QS were −20.03 mV, −21.15 mV, −25.87 mV, −26.63 mV, respectively. The addition of QS, especially 0.1~0.3%, decreased the zeta potential of each POB emulsion (pH 7.0). It was attributed to the possibility that QS with negative charge might adsorb to the POB surface. Yang et al. (2013) [25] reported that QS had carboxylic acid groups (pKa values around pH 3.5), and the groups would be fully charged (-COO-) at high pH values (pH > pKa). In addition, Liu et al. (2022) [11] reported that camellia saponins resulted in decreasing the zeta potential of camellia OB emulsions because camellia saponins and camellia OB interfacial proteins formed the complex. Under the same contents of QS, the zeta potential of the POB emulsions generally showed a decreasing trend with increasing extraction pH. The pH 4.0- and pH 5.0-POB emulsions had less charge than those of pH 6.0- and pH 7.0-POB at the same QS addition. It might be related that the initial charges of pH 4.0- and pH 5.0-POB emulsions were less (Figure 2B). Therefore, the extraction pH decreased the pIs of POBs, and the QS increased the negative charges of POB emulsions. 

### 3.3. Particle Size 

Without QS, the initial particle sizes of pH 4.0-, 5.0-, 6.0-, and 7.0-POB emulsions were 1280.50 nm, 1178.50 nm, 718.03 nm and 709.53 nm, respectively (Figure 3A). It was due to the different amounts of extrinsic proteins that were bound to POBs. It was indicated that the initial particle sizes decreased with increasing extraction pH. However, the addition of QS could decrease the particle sizes of all POB emulsions. Under the same extraction pH, the particle size decreased firstly, and then increased with increasing QS concentration from 0.05% to 0.3%. On the one hand, more QS molecules with negative charges could absorb to POB interfacial with increasing QS from 0.05% to 0.1%, which prevented droplet aggregation by increasing the electrical repulsion, which decreased the particle sizes. On the other hand, the excessive QS (more than 0.1%) might combine with POB interfacial proteins to form the complex that dissolved in the bulk phase. Li et al. (2023) [26] reported that there was an appropriate concentration of saponins combined with proteins in a certain ratio that would work best, while excessive saponins would occupy the surface layer, which affected the properties of emulsions. At the QS concentration of 0.1%, the particle sizes of pH 4.0-, 5.0-, 6.0-, and 7.0-POB emulsions were smallest, and were 774.73 nm, 620.95 nm, 341.00 nm and 398.40 nm, respectively. The particle size distributions of pH 4.0-, 5.0-, 6.0-, and 7.0-POB emulsions with 0.1% QS are shown in Figure 3B. The pH 4.0-, and 5.0-POB emulsions with 0.1% QS presented a bimodal distribution with a wide particle size distribution, while other emulsions showed a unimodal distribution. The main peak position moved toward the direction of small droplets. Therefore, the particle sizes of POB emulsions decreased with increasing extraction pH, and 0.1% QS had better effects on decreasing the particle sizes.

### 3.4. Microstructure 

The microscope images showed that some larger droplets and numerous aggregates (marked with circles in Figure 4) were obviously observed in initial pH 4.0- and 5.0-POB emulsions on the first day, while initial pH 6.0- and 7.0-POB emulsions had small and few aggregates. The aggregation was another reason for increasing the particle sizes of pH 4.0- and 5.0-POBs emulsions without QS (Figure 3A). When QS was added, the aggregates decreased and the droplets in POB emulsions were dispersed as individuals, especially pH 4.0- and 5.0-POB emulsions. This result was consistent with Figure 3, and it was due to the increase in negative charges and electrical repulsion at the presence of QS (Figure 2B). It was revealed that QS could improve the dispersion of POB emulsions. It was shown that the droplets in all emulsions with 0.1% QS was smallest and dispersed as individuals. In addition, some individual large droplets (marked with red arrows) were still observed in pH 4.0- and 5.0-POB emulsions with 0.05% and 0.3% QS. It was possible that the droplets were not completely covered with less QS, while excessive QS induced interfacial proteins released from POBs, and the QS-induced coalescence of POBs occurred. These results revealed that POB emulsions extracted at neutral pH had a better dispersion stability compared to the acid-extraction pH, and 0.1% QS had a better effect on improving dispersion stability of the POB emulsions. 

### 3.5. Storage Stability

The storage stability of emulsions is very important in food and beverage applications. The storage stability of the different POB emulsions was analyzed using macroscopic images and particle sizes during storage for 14 days. Figure 5 showed that the clear serum layer was observed in only pH 4.0-POB emulsions without QS during 14 days of storage. The creams (marked with rectangles) were observed on the top of pH 4.0-POB emulsions after 7 days of storage, and the presence of QS considerably decreased the thickness of creams. It was attributed that the droplets in pH 4.0-POB emulsions formed more and larger aggregates during storage for 7~14 days, and the addition of QS inhibited droplet aggregation. Similarly, the unclear creams (marked with rectangles) also appeared on the top of all pH 5.0-POB emulsions after 14 days of storage. However, no creams formed on the top of all pH 6.0-, and 7.0-POB emulsions during storage, which was suggested in the good dispersion. It was relative to more charges and smaller particle sizes of initial pH 6.0-, and 7.0-POB emulsions (Figure 2B and Figure 4). It can be seen that the storage stability of POB emulsions was improved with increasing extraction pH, which might be relative to the extrinsic proteins of POBs. Ishii et al. (2017) [27] reported that more extrinsic proteins of soybean OB resulted in a greater degree of particle aggregation.

Figure 6 shows that the particle sizes of all POB emulsions increased with prolonged storage days, and QS obviously decreased their particle sizes. Without QS, the particle sizes of pH 4.0-, 5.0-, 6.0-, and 7.0-POB emulsions increased to 3185.67, 2833.67, 1708.00 and 1699.50 nm after storage for 14 days, respectively. It was revealed that the higher pH was used to extract the POB, while the smaller aggregates existed in POB emulsions, which is consistent with the macroscopic images in Figure 5. In particular, the particle sizes of pH 6.0- and 7.0-POB emulsions without QS had no significant difference (*p* < 0.05) at storage for 7 and 14 days, which is attributed to the similar content and composition of POB proteins. During storage, the particle sizes of all POB emulsions obviously decreased with increasing the QS concentration from 0.05% to 0.1%, then they increased slightly with the QS increase from 0.2% to 0.3%. Under the condition of 0.1% QS, pH 6.0- and 7.0-POB emulsions had the smallest particle sizes and better storage stability than other emulsions.

### 3.6. Oxidative Stability

Oxidative stability refers to the ability of the emulsions to resist oxidative acidification and maintain the stability of the system [28]. In order to evaluate the oxidation stability of POB, the POVs (primary oxidation products) and TBARS (secondary oxidation products) were regularly monitored. Figure 7A showed that POV contents (the primary lipid oxidation product) increased first, and then decreased with prolonged storage days in all POB emulsions. It was attributed to the conversion of some primary oxidation products to secondary oxidation products [29]. In the absence of QS, the POV increased in 0~4 days, and decreased in 4~14 days, and the highest POV of pH 4.0-, 5.0-, 6.0-, and 7.0-POB emulsions were, respectively, 4.7, 4.28, 3.8, 3.93 mmol/kg at the fourth storage day. When 0.1% (*w*/*w*) QS was added into the POB emulsions, all POVs decreased, which suggested that QS could inhibit the oxidation of lipids. This might be attributed to the following aspects: first, the addition of saponins increased the amount of phenolic compounds, leading to more hydrogen donors to reduce free radicals [30]; and second, droplet charges increased by QS accumulating at the POB interface (Figure 2), which might improve the interface physical barrier and inhibit the contact between water-soluble components and lipids. As shown in Figure 7B, TBARS content of all POB emulsions gradually increased prolonged storage days at similar rates. Under the same extraction pH of POB emulsions, the TBARS of POB emulsions without QS were higher, and it was revealed QS also decreased the produce rate of secondary oxidation products. In addition, the POV and TBARS in pH 6.0- and 7.0-POB emulsions were lower than those of pH 4.0- and 5.0-POB emulsions at the same concentration of QS, which might be because pH 6.0- and 7.0-POB emulsions had more charges (Figure 2) and less extrinsic proteins (Figure 1B). Zhao et al. (2016) [15] reported that extrinsic proteins of soybean OBs played a crucial role in accelerating the oxidation of lipids, and oxidative stability increased with decreasing extrinsic proteins. Therefore, QS (0.1%, *w*/*w*) improved the oxidation stability of POB emulsions, and POB emulsions extracted at higher pH had better oxidation stability.

## 4. Conclusions 

In this study, we prepared emulsions with POBs extracted at pH 4.0, 5.0, 6.0, 7.0, and discussed the effects of QS on physicochemical properties of POB emulsions. It was found that moisture and protein contents decreased, while lipid content increased with increasing extraction pH. All POBs had 15 kDa oleosin and 17 kDa oleosin, while the different POBs had different extrinsic proteins. At the lower extraction pH (4.0 and 5.0), extrinsic proteins (23 kDa and 38 kDa glycoproteins, unknown proteins 48 kDa and 60 kDa) were bounded to POBs, while 48 kDa and 38 kDa were completely removed at extraction pH 6.0 and 7.0. The particle sizes and zeta potential of POB emulsions decreased, whereas pI, storage stability and oxidative stability increased in the order of pH 4.0-, 5.0-, 6.0- and 7.0-POB. QS (0.05~0.3%, *w*/*w*) increased the negative charges of POB emulsions, decreased the particle sizes, and inhibited the droplet aggregation and oxidation of lipids. 0.1% QS could significantly improve the dispersion, storage and oxidative stability of POB emulsions. Therefore, extraction pH and QS had important effects on the physicochemical properties of POB emulsions. It was considered that the pH 6.0- and 7.0-POB emulsions with 0.1% QS could be used for plant beverages. This study will provide a strategy for improving the stability of OB emulsions and the application of POBs in the food industry. 

## Figures and Tables

**Figure 1 foods-12-03017-f001:**
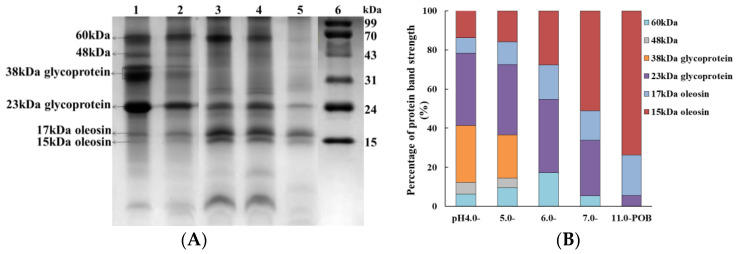
(**A**) Tricine-SDS-PAGE of POBs extracted at different pH. Lanes 1~5, pH 4.0-, 5.0-, 6.0-, 7.0- and 11.0-POB; Lane 6, protein marker. (**B**) Percentages of 60 kDa, 48 kDa, glycoprotein 38 kDa, glycoprotein 23 kDa, and oleosins in total proteins of pH 4.0-, 5.0-, 6.0-, 7.0- and 11.0-POB.

**Figure 2 foods-12-03017-f002:**
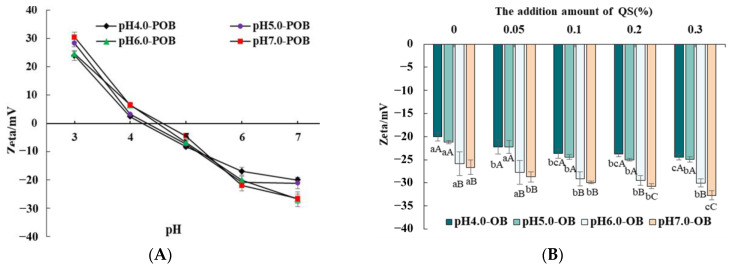
(**A**) Zeta potential of pH 4.0-, 5.0-, 6.0-, and 7.0-POBs at the different environment pH. (**B**) Effect of QS (0~0.3%, *w*/*w*) on the zeta potential of pH 4.0-, 5.0-, 6.0-, and 7.0-POB emulsions. Different capital or small letters represent significant differences (*p* < 0.05).

**Figure 3 foods-12-03017-f003:**
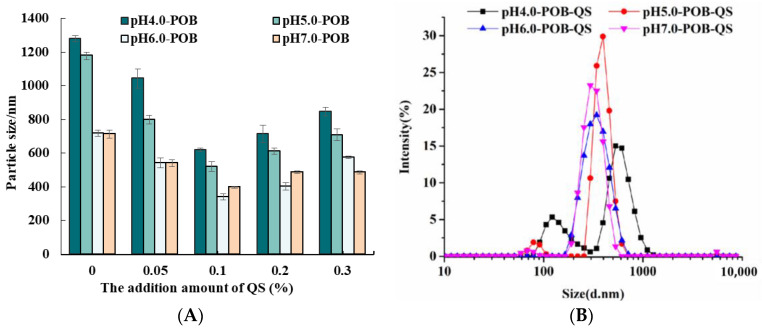
(**A**) Effect of QS (0~0.3%, *w*/*w*) on the particle size of pH 4.0-, 5.0-, 6.0-, and 7.0-POB emulsions. (**B**) The size distributions of pH 4.0-, 5.0-, 6.0-, and 7.0-POB emulsions with 0.1% (*w*/*w*) QS.

**Figure 4 foods-12-03017-f004:**
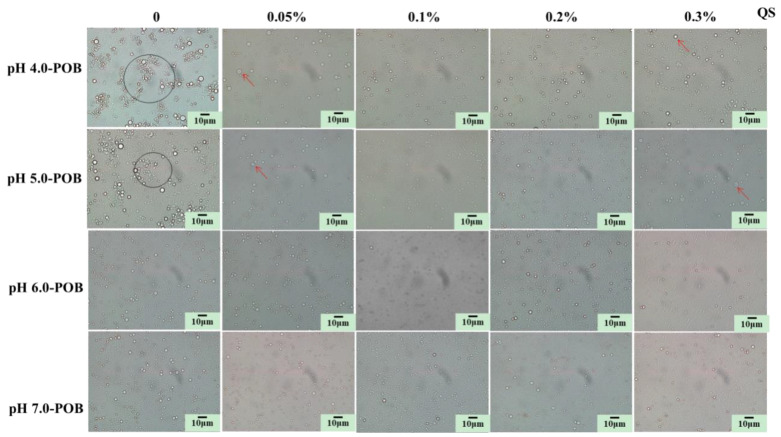
Effect of QS (0~0.3%, *w*/*w*) on the microstructure of pH 4.0-, 5.0-, 6.0-, and 7.0-POB emulsions. The larger droplet aggregates were marked by circles, and single large droplets were marked by red arrows.

**Figure 5 foods-12-03017-f005:**
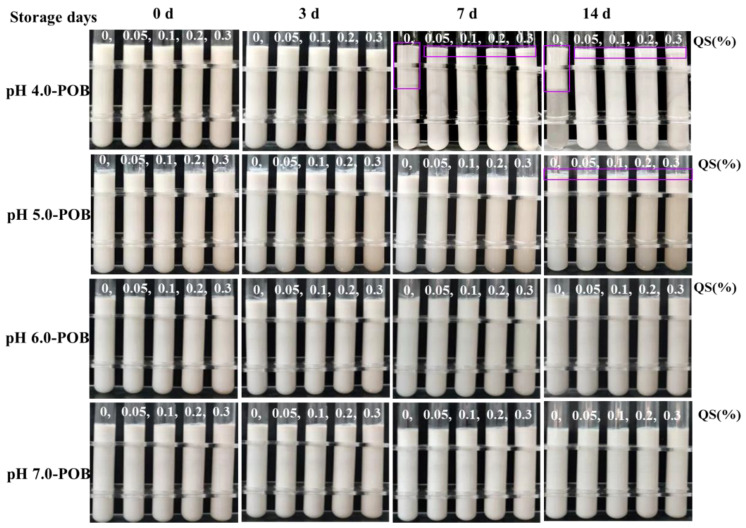
Effect of QS (0~0.3%, *w*/*w*) on the visual appearance of pH 4.0-, 5.0-, 6.0-, and 7.0-POB emulsions during storage for 0, 3, 7, 14 days. Creams were marked by rectangles.

**Figure 6 foods-12-03017-f006:**
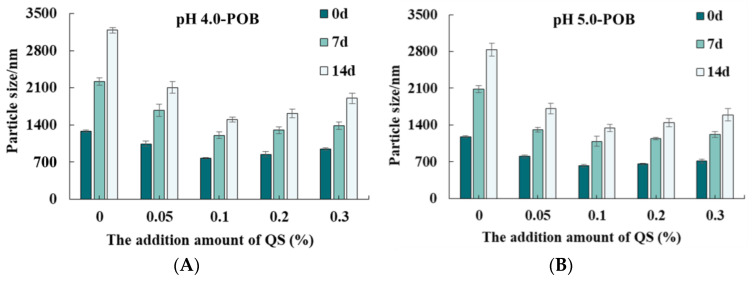
Effect of QS content (0~0.3%, *w*/*w*) on the particle size of pH 4.0-(**A**), 5.0-(**B**), 6.0-(**C**), and 7.0-POB emulsions (**D**) during storage.

**Figure 7 foods-12-03017-f007:**
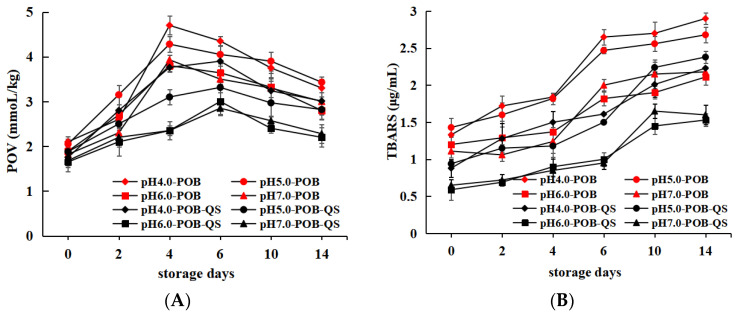
POV (**A**) and TBARS (**B**) of pH 4.0-, 5.0-, 6.0- and 7.0-POB emulsions without QS and with 0.1% (*w*/*w*) QS as a function of storage days.

**Table 1 foods-12-03017-t001:** Contents of moisture, protein, and lipid of pH 4.0-, 5.0-, 6.0-, and 7.0-POB. Different small letters represent significant differences (*p* < 0.05) in the same component.

Samples	Compositions		
	Moisture (%)	Lipid (%, Dry Basis)	Protein (%, Dry Basis)
pH 4.0-POB	52.39 ± 0.69 ^a^	57.60 ± 3.45 ^c^	33.40 ± 0.37 ^a^
pH 5.0-POB	46.14 ± 1.30 ^b^	76.76 ± 0.93 ^b^	13.57 ± 1.49 ^b^
pH 6.0-POB	28.75 ± 1.07 ^c^	83.52 ± 2.18 ^ab^	4.82 ± 0.13 ^c^
pH 7.0-POB	20.98 ± 1.25 ^d^	86.31 ± 4.16 ^a^	4.27 ± 0.54 ^c^

## Data Availability

The data presented in this study are available on request from the corresponding author.

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
