# Peer review of "Effects of Quillaja Saponin on Physicochemical Properties of Oil Bodies Recovered from Peony (Paeonia ostii) Seed Aqueous Extract at Different pH"

_foods, 2023, doi:10.3390/foods12163017_

Round 1

Reviewer 1 Report

Title: Effects of Quillaja Saponin on Physicochemical Properties of Oil Bodies Recovered from Peony (Paeonia ostii) Seed Aqueous Extract at Different pHs.

Overview and general comments:

The manuscript deals with the effect of Quillaja saponin on the physicochemical properties of oil bodies recovered from peony seed aqueous extract. The results are interesting and the subject is within the scope of the journal. However, I have several comments to be considered regarding mainly the language of the manuscript and the current presentation of the obtained results. Indeed, the text should be well checked by a native English speaker. Simple, clear, short and “grammatically” correct sentences are highly recommended. I could not easily understand several parts of the manuscript due to the written language and it was quite overwhelming to distinct between authors’ point of views and previously reported explanations.

The guideline of the journal should also be respected and authors must carefully follow the instructions. For instance, the authors should insert their graphics (schemes, figures, etc.) in the main text after the paragraph of its first citation. Author Contributions section should also be included in the manuscript.

Specific comments:

Starting from the title, Quillaja should be written in italic. This should also be corrected in the whole manuscript.

For the abstract, the authors should re-write the abstract in a more convenient manner. They indeed reported the results in the abstract without giving insights on the used methodology. They should start with the objective of the study; the methodology (methods/approach) and then the results. According to the journal’s guideline: ”the abstract should be a single paragraph and should follow the style of structured abstracts, but without headings: 1) Background: Place the question addressed in a broad context and highlight the purpose of the study; 2) Methods: Describe briefly the main methods or treatments applied. Include any relevant preregistration numbers, and species and strains of any animals used; 3) Results: Summarize the article's main findings; and 4) Conclusion: Indicate the main conclusions or interpretations”. The abstract in its current presentation seems to be a brief presentation of the obtained results.

A special attention should also be dedicated to the abbreviation as some abbreviations were not defined (QS: Quillaja saponin). Description of the obtained extracts as “pH 4-POB” for example is overwhelming and thus needs to be presented in a better manner.

Concerning the POBs extracted at different pHs, the description should include more details. For instance, after pH adjustment did you recover the precipitate or the liquid phase? At pH4, I believe that you have had a precipitate. Basically you mentioned that the floating parts were collected. However, since you have obtained a decreasing protein content while increasing the pH, this seems quite confusing as the proteins should be in the precipitate. Thus, while collecting the upper floats, the protein content should increase with pHs!

The section 2.9.1 lipid hydroperoxydes: please re-write this section while using the past tense.

Line 154: “Table 1 showed that the main compositions of the POBs extracted at pH 4.0~7.0.” please correct.

Line 182: “It was suggested that the interaction forces between different extrinsic proteins and POBs were different”. Please re-write this and add the appropriate reference.

Line 196: “It was suggested that the interaction forces between different extrinsic proteins and POBs were different”. Please re-write this and add the appropriate reference.

Lines 199-200: “It was attributed that QS with negative charge adsorbed to POB interfacial proteins” Please re-write this and add the appropriate reference.

Lines 205-206: “It might be related that the initial charges of pH 4.0- and pH 5.0-POB emulsions were less”. Please correct the sentence and add the appropriate reference.

Considering the use of “%” sometimes the authors use a space between values and “%”, sometimes not. Please use similar presentation of the obtained values in all the manuscript.

For the conclusion section, instead of focusing on rewriting the obtained results I suggest the authors to rewrite it while presenting the results briefly and focusing on the importance of the findings and eventually the future perspectives.

Reviewer 2 Report

The paper is very interesting and is based on valuable experimental data for valorization of new resources   for food industry.

Presentation and general writing have to be improved because some of the phrases are too short and difficult to follow .

Moreover even if the use of the Journal word template is not compulsory, the reading of the paper would have highly benefit from using it.

There are some spelling  and formulation errors for example in  conclusions , line 302 the phrase does not reflect the data in table 1.

I wonder why was sodium azide used as preservative if the intended use of the emulsions obtained from SOB is foods?

A thorough English editing is required because the paper is difficult to  be read.

Reviewer 3 Report

Please see the file.

Round 2

Reviewer 1 Report

After making revisions to the original version submitted for evaluation, the article has undergone significant improvements. Therefore, I strongly advise publishing the article in its current enhanced state.

Author Response

Response to Reviewer 1 Comments

Point 1: Are all the cited references relevant to the research? Can be improved.

Response 1: we have been checked manuscript and all references are relevant to this study. Some changes are highlighted with a red font.
